# Computer Model of Synapse Loss During an Alzheimer’s Disease-Like Pathology in Hippocampal Subregions DG, CA3 and CA1—The Way to Chaos and Information Transfer

**DOI:** 10.3390/e21040408

**Published:** 2019-04-17

**Authors:** Dariusz Świetlik, Jacek Białowąs, Janusz Moryś, Aida Kusiak

**Affiliations:** 1Intrafaculty College of Medical Informatics and Biostatistics, Medical University of Gdańsk, 1 Debinki St., 80-211 Gdańsk, Poland; 2Department of Anatomy and Neurobiology, Medical University of Gdańsk, 1 Debinki St., 80-211 Gdańsk, Poland; 3Department of Periodontology and Oral Mucosa Diseases, Medical University of Gdańsk, 1a Debowa St., 80-204 Gdańsk, Poland

**Keywords:** neural networks, Alzheimer’s disease, learning and memory, hippocampus, LTP, theta rhythm, computer simulation

## Abstract

The aim of the study was to compare the computer model of synaptic breakdown in an Alzheimer’s disease-like pathology in the dentate gyrus (DG), CA3 and CA1 regions of the hippocampus with a control model using neuronal parameters and methods describing the complexity of the system, such as the correlative dimension, Shannon entropy and positive maximal Lyapunov exponent. The model of synaptic breakdown (from 13% to 50%) in the hippocampus modeling the dynamics of an Alzheimer’s disease-like pathology was simulated. Modeling consisted in turning off one after the other EC2 connections and connections from the dentate gyrus on the CA3 pyramidal neurons. The pathological model of synaptic disintegration was compared to a control. The larger synaptic breakdown was associated with a statistically significant decrease in the number of spikes (R = −0.79, *P* < 0.001), spikes per burst (R = −0.76, *P* < 0.001) and burst duration (R = −0.83, *P* < 0.001) and an increase in the inter-burst interval (R = 0.85, *P* < 0.001) in DG-CA3-CA1. The positive maximal Lyapunov exponent in the control model was negative, but in the pathological model had a positive value of DG-CA3-CA1. A statistically significant decrease of Shannon entropy with the direction of information flow DG->CA3->CA1 (R = −0.79, *P* < 0.001) in the pathological model and a statistically significant increase with greater synaptic breakdown (R = 0.24, *P* < 0.05) of the CA3-CA1 region was obtained. The reduction of entropy transfer for DG->CA3 at the level of synaptic breakdown of 35% was 35%, compared with the control. Entropy transfer for CA3->CA1 at the level of synaptic breakdown of 35% increased to 95% relative to the control. The synaptic breakdown model in an Alzheimer’s disease-like pathology in DG-CA3-CA1 exhibits chaotic features as opposed to the control. Synaptic breakdown in which an increase of Shannon entropy is observed indicates an irreversible process of Alzheimer’s disease. The increase in synapse loss resulted in decreased information flow and entropy transfer in DG->CA3, and at the same time a strong increase in CA3->CA1.

## 1. Introduction

One of the most current issues in medicine is a detailed explanation of causes, possibilities of diagnosis and treatment of Alzheimer’s disease (AD), which causes in the late stage a complete loss of memory regarding current events and changes in the environment [1]. Connections between the entorhinal cortex and hippocampus are essential for this disease. At a late stage, these connections are completely destroyed. In the initial phase, only a few out of thousands of connections from the stellate cells of the second layer of the entorhinal cortex are discontinued, but this always leads to a decrease in the possibility of learning and memorizing new changes in the environment. This phenomenon can be easily modeled even in simplified models of the hippocampus network. In AD, the hippocampus plays a special role, which is important in impaired memory functioning [2]. The experiments show that a 50% decrease in the number of synaptic connections is the main and most credible factor of cognitive deficiency, which is called synaptic deletion [2]. Brain compensation mechanisms include the strengthening of the remaining synaptic connections, which are referred to as synaptic compensations. Models of synaptic breakdown and compensation are based on these observations [3,4,5]. According to these models, it has been proven that in the Hopfield network architecture, the loss of synaptic connections is the cause of memory decline and distortion of learned patterns. However, uncontrolled synaptic modification is another phenomenon seen in the associative network [6,7,8].

Models of neural networks in experiments with brain simulations carried out to understand the basic mechanisms of AD can be divided into types of so-called connection or biophysical models. Models of neurons used in simulation systems of processes associated with associative memory must demonstrate the ability of time-space integration [2,9,10,11]. Models of neural networks are used in simulations of memory dysfunctions, such as AD, where theta and gamma oscillation is required, as well as various types of neurons and input patterns [12,13,14].

The most commonly used methods characterizing the complexity of the system that is the hippocampus are the positive maximal Lyapunov exponent, correlative dimension and Shannon entropy [15,16]. The study [15] showed that patients with AD were characterized by significantly higher values of Lyapunov exponents and correlative dimensions in EEG studies compared with controls.

Entropy is a concept referring to randomness and predictability, therefore, greater entropy is often associated with greater randomness and smaller systemic order [17]. We have two families of entropy estimators: spectral entropy and entropy deposition [18]. The usefulness of entropy in the analysis of EEG signals in AD has been demonstrated. The increased value of entropy in AD has been shown in previous studies [17,18,19,20,21].

The aim of this study was to model the synaptic breakdown in AD using methods describing the complexity of systems, such as Lyapunov entropy or correlation dimension. Our experiment with simulations of the hippocampus network DG-CA3-CA1 was aimed at understanding the basic mechanisms of AD and information flow in the hippocampus.

## 2. Materials and Methods

### 2.1. Study Design

Five simulations were performed: one of them was characterized by a control model (control model), and another four a pathological model (pathological model). Figure 1 presents a detailed diagram of the simulation of the hippocampus network DG-CA3-CA1. The pathological model simulated synaptic breakdown in the hippocampus for four consecutive phases: 9%, 18%, 26% and 35% of the synapse loss modeling the dynamics of AD. Modeling of AD consisted in turning off one after the other connections with EC2 and connections with EC2 on inhibitory interneurons on the granule cells of the dentate gyrus and on CA3 pyramidal neurons (basket cells within the dentate gyrus (DG) and CA3) (broken connections are marked red on Figure 1). In our parallel simulation study, we performed a comparative analysis of neuronal parameters, the complexity of the hippocampus through a positive Lyapunov exponent, correlative dimension and Shannon entropy and parameters of information theory (entropy transfer and mutual information).

### 2.2. The Model Description

The mathematical formalism of the DG-CA3-CA1 network was based on previous studies [22,23,24]. The diagram of the neural network related to the hippocampal subregions DG-CA3-CA1 is shown in Figure 2. Our model was built of 21 cells. The DG region contained four granule cells and three inhibitory interneurons: two basket cells and one mossy cell (O-LM). On the other hand, the CA3 and CA1 areas contained four pyramidal cells and three inhibitory interneurons: two basket cells and one O-LM cell. The simplified morphology of nerve cells including the cell body, part of the axon and dendrites (mainly apical) was used. All the properties of the nerve cell used in the experiment were based on the functions described in the literature [25,26,27,28,29,30].

### 2.3. Synaptic Properties

All cell models—pyramid, basket and O-LM—were made of 16 compartments. Each dendrite had excitatory or inhibitory synapses. The mathematical formalism describing AMPA, NMDA and GABA receptors from previous works was used [22,23,24]. Each CA3 pyramidal cell received inhibitory synapses from basket cells and O-LM cells. The inputs stimulating the CA3 pyramidal cells were received from the second layer of entorhinal cortex and dentate gyrus. On the other hand, stimulating input basket cells were received from the distal dendrites on the second layer of the entorhinal cortex and from the dentate gyrus of mossy fiber cells. The inputs from EC2 and EC3 were shifted in phase relative to each other, so that the strong stimulation from one corresponded to the weak stimulation of the other [31]. Each O-LM cell had stimulation inputs from CA3 pyramidal cells and an inhibition from the septum [32,33]. According to biological research, the sources of inputs to the CA1 area were mossy fibers and Schaffer’s collaterals from CA3 and projections from the third layer of the entorhinal cortex [34]. In addition to this, theta oscillation was delivered via the septo-hippocampal system through the vault and described frequencies in the band from 4 Hz to 12 Hz, and temporarily anchored in faster gamma oscillations, which was simulated in this study [35,36,37,38,39]. It is believed that theta oscillations play a fundamental role in the activity of the hippocampus, including spatial information [40,41,42].

### 2.4. Correlation Dimension, Shannon Entropy and the Positive Maximal Lyapunov Exponent

For nonlinear analysis of the results of the pathological and the control model, phase space reconstruction was performed, which is one of the methods used for describing the complexity of the dynamic system [43]. Attractor reconstruction was performed using time delay methods [44,45]. The mutual information method was used to determine the ‘optimal’ time delay value for the reconstruction of the state space [46]. The minimum dimension of deposition of one-dimensional time series was chosen using the false nearest neighbors method [47]. Finally, the correlation dimension, Shannon entropy and the positive maximal Lyapunov exponent were established with the use of a tool proposed by Charles Webber and Joseph Zbilut (recurrence quantification analysis) [48]. In 1948, Claude Shannon defined entropy as a measure of uncertainty associated with a random variable. The Shannon entropy measures the recursion power of the studied time series, which in some ways allows the ability to check the degree of their chaoticity [49].

### 2.5. Mutual Information and Transfer Entropy

Mutual information (MI) is a concept in the field of information theory that defines the relationship between two random variables and can be used as an alternative to a well-known correlation analysis [50,51]. Therefore, MI can help more than correlation analysis to understand the interaction between the two systems and determine the degree of coupling of the two systems [52]. Mutual information measures how much information we can have about the X signal, knowing Y, but does not provide knowledge about the dynamics and direction of its flow. This problem is solved by using the entropy transfer method [52]. The use of the entropy transfer method in the analysis of interactions between two systems can allow us to distinguish information that has really been exchanged between the systems from information that is the same in both systems due to a common source of information or a common history. The entropy transfer can determine the information exchanged between the systems separately for each of the directions.

### 2.6. Statistical Methods and Software

The statistical analyses were performed using the statistical suite TIBCO Software Inc. (2017), Statistica (data analysis software system), version 13, (Palo alto, CA, USA, 2017, http://statistica.io) and Excel. The significance of difference between more than two groups was assessed with a parametric F test (ANOVA). In case of statistically significant differences between two groups, post-hoc tests were used (Tukey test). Two-way ANOVA with post-hock Tukey analysis were performed using TIBCO Software Inc. (2017), Statistica. Chi-squared tests for independence were used for qualitative variables. In order to determine dependence, strength and direction between variables, correlation analysis was used by determining the Pearson or Spearman’s correlation coefficients. In all the calculations, the statistical significance level of *P* < 0.05 has been used. Parameter calculations for complex systems and information theory were made in the Neuroscience Information Theory Toolbox software [53].

## 3. Results

### 3.1. Neuronal Parameters

The performed simulations compared the number of spikes, spikes per burst, burst duration and inter-burst interval in a control model with pathological simulations associated with increasing synaptic breakdown from 9% to 35%.

#### 3.1.1. DG-CA3-CA1 and CA3-CA1 Areas

Statistical analyses of the results of number of spikes in the hippocampal cumulative areas DG-CA3-CA1 show statistically significant differences between the pathological and control model. A statistically significant reduction in the number of spikes was obtained in the synaptic breakdown model at the 26% and 35% levels compared with the control (268.4 and 288.1 vs. 504.8, *P* < 0.001). In addition, the number of spikes was significantly lower in the synaptic disintegration model at 26% and 35% compared with 9% (268.4 and 288.1 vs. 412.9, *P* < 0.01). A statistically significant decrease in the number of spikes was also obtained at 26% and 35% compared with 18% (268.4 and 288.1 vs. 399.3, *P* < 0.05) (Figure 3). A statistically significant negative correlation in the level of synaptic decay was obtained, and in the number of spikes (correlation coefficient R = −0.79, *P* < 0.001).

Similarly, in the area of CA3-CA1, statistically significant differences were obtained between the pathological model and the control model. There was a statistically significant reduction in the number of spikes in the synaptic breakdown model at the 26% and 35% levels compared with the control (286.6 and 315.1 vs. 550.6, *P* < 0.001). In addition, the number of spikes was significantly smaller in the synaptic breakdown model at the 26% level compared to the 9% level (286.6 vs. 442.9, *P* < 0.01) (Figure 3). A statistically significant negative correlation was obtained in the level of synaptic breakdown, and in the number of spikes (correlation coefficient R = −0.83, *P* < 0.001).

Statistically significant differences were obtained for spikes per burst by comparing the pathological model with the control. There was a statistically significant reduction in spikes per burst in the synaptic breakdown model at the 26% and 35% levels compared with the control (2.5 and 2.8 vs. 5.4, *P* < 0.001). In addition, spikes per burst was significantly lower in the synaptic breakdown model at the 26% and 35% levels compared with the 9% level (2.5 and 2.8 vs. 4.2, *P* < 0.01). A statistically significant drop in spikes per burst was also obtained at the 26% and 35% levels compared with the breakdown at the 18% level (2.5 and 2.8 vs. 4.1, *P* < 0.05) (Figure 3). A statistically significant negative correlation in the level of synaptic breakdown was obtained, and in the spikes per burst (correlation coefficient R = −0.76, *P* < 0.001).

The spikes per burst analysis was statistically significant in the synaptic breakdown model at the 26% and 35% levels compared with the control (2.8 and 3.2 vs. 6.0, *P* < 0.001) for the CA3-CA1 region. In addition, spikes per burst was significantly lower in the synaptic disintegration model at the 26% and 35% levels compared with the 9% level (2.8 and 3.2 vs. 4.6, *P* < 0.01) (Figure 3). A statistically significant negative correlation in the level of synaptic breakdown was obtained, and in spikes per burst (correlation coefficient R = −0.81, *P* < 0.001).

Statistical analysis of burst duration showed a statistically significant reduction in time in the synaptic breakdown model at the level of 9%, 26% and 35% compared with the control (33.3, 18.6 and 19.4 vs. 41.9, *P* < 0.05, *P* < 0.001) in DG-CA3-CA1. In addition to this, shortened burst duration at the breakdown level of 26% compared to the 9% level (18.6 vs. 33.3, *P* < 0.01) and at the 26% and 35% level compared to 18% level (18.6 and 19.4 vs. 35.2, *P* < 0.01) was obtained (Figure 3). A statistically significant negative correlation was obtained in the level of synaptic breakdown and burst duration (correlation coefficient R = −0.83, *P* < 0.001).

In the CA3-CA1 model, statistical analysis of the burst duration showed a statistically significant reduction of time in the synaptic breakdown model at the 26% and 35% level compared with the control (18.1 and 19.1 vs. 42.1, *P* < 0.001). The time reduction was also obtained at the breakdown level of 38% compared with the 9% level (18.1 vs. 33.1, *P* < 0.01) (Figure 3). A statistically significant negative correlation was obtained at the level of the synaptic breakdown and the burst duration (correlation coefficient R = −0.83, *P* < 0.001).

The results of the statistical inter-burst interval analyses showed a statistically significant increase in the 9%, 26% and 35% models compared with the control (93.0, 112.0 and 111.8 vs. 84.1, *P* < 0.05, *P* < 0.001) in DG-CA3-CA1. In addition to this, an increase in the inter-burst interval was obtained in the 38% and 50% models compared with the 9% and 18% models (112.0 and 111.8 vs. 93.0 and 91.5, *P* < 0.01) (Figure 3). A statistically significant positive correlation in the level of synaptic decay was obtained, and in the inter-burst interval (correlation coefficient R = 0.85, *P* < 0.001).

The results of the statistical inter-burst interval analysis showed a statistically significant increase in the 9%, 26% and 35% models compared with the control (93.9, 113.1 and 112.9 vs. 84.3, *P* < 0.001) in the CA3-CA1 area. In addition to this, an increase in the inter-burst interval was obtained in the 26% and 35% models compared with the 25% model (113.1 and 112.9 vs. 92.8, *P* < 0.05) (Figure 3). A statistically significant positive correlation was obtained at the level of synaptic breakdown, and the inter-burst interval (correlation coefficient R = 0.86, *P* < 0.001).

#### 3.1.2. DG, CA3 and CA1 Regions

Statistical analyses of the results of the number of spikes in the hippocampal accumulation of DG, CA3 and CA1 areas also show statistically significant differences between the pathological model and the control model. A statistically significant reduction in the number of spikes was obtained in the synaptic breakdown model at the 26% and 35% level compared with the control (232.0 and 234.0 vs. 413.0, *P* < 0.05) in DG. In addition to this, the number of spikes was significantly smaller at the 26% synaptic breakdown model level compared with the 9% level (232.0 vs. 353.0, *P* < 0.05) in DG (Figure 4). In the CA3 model, a statistically significant reduction in the number of spikes in the synaptic breakdown model was obtained at the 26% and 35% levels compared with the control (300.0 and 329.3 vs. 587.8, *P* < 0.05). In contrast, in the CA1 region, a significant decrease in the number of spikes was obtained in the synaptic breakdown model at the 26% level compared with the control (273.3 vs. 513.5, *P* < 0.05) (Figure 4). There were statistically significant negative correlations at the level of synaptic breakdown, and number of spikes for DG (correlation coefficient R = −0.80, *P* < 0.001), CA3 (correlation coefficient R = −0.86, *P* < 0.001) and CA1 (correlation coefficient R = −0.82, *P* < 0.001).

The spikes per burst analysis was statistically significant in the synaptic breakdown model at the 26% and 35% levels compared with the control (2.0 and 2.0 vs. 4.2, *P* < 0.05) for DG. In addition to this, the spikes per burst was significantly smaller in the synaptic breakdown model at the 26% level compared with the 18% level (2.0 vs. 3.7, *P* < 0.05) for DG (Figure 3). In the CA3 model, a statistically significant reduction in spikes per burst was obtained in the synaptic breakdown model at the 26% and 35% levels compared with the control (2.9 and 3.3 vs. 6.4, *P* < 0.05). In contrast, in the CA1 region, a significant decrease in the spikes per burst was obtained in the model of synaptic breakdown at the 26% level compared with the control (2.6 vs. 5.5, *P* < 0.05) (Figure 4). Statistically significant negative correlations were obtained at the level of synaptic breakdown, and of spikes per burst for DG (correlation coefficient R = −0.80, *P* < 0.001), CA3 (correlation coefficient R = −0.85, *P* < 0.001) and CA1 (correlation coefficient R = −0.79, *P* < 0.001).

On the other hand, the statistical analysis of burst duration showed a statistically significant reduction of time in the synaptic breakdown model at the 26% and 35% levels compared with the control (19.6 and 19.9 vs. 41.6, *P* < 0.05) in DG. The time reduction was also obtained at the 26% breakdown level compared with the 18% level (19.6 vs. 36.0, *P* < 0.05) in DG (Figure 4). In the CA3 model, a statistically significant shortening of burst duration was obtained in the synaptic breakdown model at the 26% and 35% levels compared with the control (18.7 and 20 vs. 43.0, *P* < 0.05). In contrast, in the CA1 region, a significant shortening of burst duration was obtained in the synaptic breakdown model at the 26% and 35% levels compared with the control (17.5 and 18.2 vs. 41.2, *P* < 0.05) (Figure 4). Statistically significant negative correlations at the level of synaptic breakdown were obtained, and a burst duration for DG (correlation coefficient R = −0.80, *P* < 0.001), CA3 (correlation coefficient R = −0.85, *P* < 0.001) and CA1 (correlation coefficient R = −0.85, *P* < 0.001).

The results of the statistical inter-burst interval analyses showed a statistically significant increase in the 26% and 35% models compared with the control (110.0 and 109.7 vs. 83.7, *P* < 0.05) in DG. In addition to this, an increase in the inter-burst interval was obtained in the 26% model compared with the 18% model (110.0 vs. 88.7, *P* < 0.05) in DG (Figure 4). In the CA3 model, a statistically significant increase in the inter-burst interval was obtained in the synaptic breakdown model at the 26% and 35% levels compared with the control (110.9 and 111.3 vs. 83.4, *P* < 0.05). In the CA1 region, however, a significant increase in the inter-burst interval was obtained in the synaptic breakdown model at the 26% and 35% levels compared with the control (115.2 and 114.5 vs. 85.2, *P* < 0.05) (Figure 4). Statistically significant positive correlations were obtained at the level of synaptic breakdown and burst duration for DG (correlation coefficient R = 0.80, *P* < 0.001), CA3 (correlation coefficient R = 0.85, *P* < 0.001) and CA1 (correlation coefficient R = 0.84, *P* < 0.001).

### 3.2. Parameters in a Complex System: Hippocampus

In simulations of correlation dimension, Shannon entropy and the positive maximal Lyapunov exponent in the control compared with the pathological model in simulations related to the increasing synaptic disintegration from 9% to 35%.

#### 3.2.1. DG-CA3-CA1 and CA3-CA1 Regions

Statistical analyses of the correlation dimension results in hippocampal simulations of the DG-CA3-CA1 regions show statistically significant differences between the pathological and the control model. A statistically significant decrease in the correlation dimension was obtained in the synaptic breakdown model at the 35% level compared with the control (3.9 vs. 6.0, *P* < 0.05). In addition to this, a statistically significant decrease in the correlation dimension was found in the synaptic breakdown model at the 35% level compared to synaptic breakdown at the 26% level (3.9 vs. 6.6, *P* < 0.01).

However, in the CA3-CA1 region there were no statistically significant differences regarding the correlation dimension between the tested models (*P* = 0.8452) (Figure 5).

The results of the statistical analyses of Shannon entropy changes relative to the synaptic breakdown in the DG-CA3-CA1 model also did not show statistical significance (*P* = 0.3528) (Figure 4). On the other hand, in the CA3-CA1 region a statistically significant increase in Shannon entropy in the synaptic breakdown model at the 26% and 35% levels was obtained as compared with the control (1.9 and 1.6 vs. 1.0, *P* < 0.05) (Figure 5).

Detailed statistical analysis showed statistically significant differences in the positive maximal Lyapunov exponent relative to the synaptic breakdown in the DG-CA3-CA1 and CA3-CA1 regions. In the DG-CA3-CA1 region a statistically significant increase in the positive maximal Lyapunov exponent was obtained at all levels of synaptic breakdown compared with the control model (0.053, 0.048, 0.036 and 0.034 vs. −0.136, *P* < 0.01, *P* < 0.05) (Figure 5). Similar results were obtained in the CA3-CA1 region (0.066, 0.060, 0.042 and 0.042 vs. −0.213, *P* < 0.01, *P* < 0.05) (Figure 5).

#### 3.2.2. DG, CA3 and CA1 Regions

Statistical analyses of the correlation dimension results in the hippocampal simulations of the DG, CA3 and CA1 regions show statistically significant differences between the pathological and the control model only in the DG region. A statistically significant decrease in the correlation dimension was obtained in a synaptic breakdown model at the 35% level compared with the control (1.0 vs. 7.0, *P* < 0.05). In addition to this, a statistically significant increase in the synaptic breakdown model at the 26% and 35% levels was obtained as compared with the 26% level (5.0 and 1.0 vs. 10.0, *P* < 0.01). There were no statistically significant differences between the correlation dimension and the synaptic breakdown in the CA3 and CA1 regions, (*P* > 0.05) (Figure 6).

The results of the statistical analyses showed statistically significant Shannon entropy changes relative to the synaptic breakdown in the DG and CA1 regions. Shannon entropy significantly increased at the 35% level compared with the control model (3.5 vs. 2.8, *P* < 0.01) and at the 18% level compared to the 35% level (3.0 vs. 3.5, *P* < 0.05) in the DG region. In contrast, in the CA1 region a statistically significant increase in Shannon entropy in the synaptic breakdown model at the 35% level was obtained compared with the control (1.8 vs. 0.4, *P* < 0.01). In addition to this, a statistically significant increase at the 9% and 18% levels was obtained compared with the 35% level (0.8 and 0.6 vs. 1.8, *P* < 0.5, *P* < 0.01) (Figure 6). A statistically significant increase in Shannon entropy in the synaptic breakdown model at the 9% and 18% levels was obtained compared with the 26% level (0.8 and 0.6 vs. 1.9, *P* < 0.001) (Figure 6).

A detailed statistical analysis showed statistically significant differences in the positive maximal Lyapunov exponent relative to the synaptic breakdown in the DG, CA3 and CA1 regions. In the DG region, a statistically significant increase in the positive maximal Lyapunov exponent at the synaptic breakdown at the 9% and 18% levels relative to the control model (0.027 and 0.025 vs. 0.018, *P* < 0.5, *P* < 0.001) was obtained (Figure 6). On the other hand, at the 26% and 35% levels a significantly lower value was obtained with respect to the 9% level (0.024 and 0.020 vs. 0.027, *P* < 0.05, *P* < 0.001) (Figure 6).

In the CA3 region a statistically significant increase in the positive maximal Lyapunov exponent at all levels of synaptic breakdown compared with the control model (0.060, 0.035, 0.045 and 0.036 vs. −0.230, *P* < 0.05) was obtained (Figure 6).

Similarly, in the CA1 region, a statistically significant increase in the positive maximal Lyapunov exponent at all levels of synaptic breakdown compared with the control model (0.072, 0.085, 0.039 and 0.047 vs. −0.197, *P* < 0.05) was obtained (Figure 6).

### 3.3. The Flow of Information in Hippocampus *vs.* Shannon Entropy, Transfer Entropy and Mutual Information

The two-factor ANOVA analysis of three hippocampal regions (DG->CA3->CA1 information flow) showed a statistically significant decrease in Shannon entropy with the information flow direction (*P* < 0.001) and a statistically significant increase with greater synaptic breakdown (*P* < 0.01). The value of Shannon entropy (*P* < 0.001) decreased both in the pathological and control model together with the information flow direction (Figure 7).

In the DG region, Shannon entropy also increased with the increase of synaptic breakdown (correlation coefficient R = 0.49, *P* < 0.01). A similar correlation was obtained for the CA1 region (correlation coefficient R = 0.48, *P* < 0.05) and the CA3-CA1 region (correlation coefficient R = 0.24, *P* < 0.05). In the CA3 and DG-CA3-CA1 regions, however, statistically significant relations between synaptic breakdown and Shannon entropy were not obtained (Figure 7).

When analyzing synaptic degradation, a decrease in mutual information (MI) in DG->CA3 and an increase in CA3->CA1 is shown. The increase in synaptic degradation caused a decrease in transfer entropy (TE) for DG->CA3. In contrast, for CA3->CA1, the inverse relationship was obtained (Figure 7). Transfer entropy for DG->CA3 decreased to 35% relative to the control at the synaptic breakdown level of 35%. In addition to this, the same relationship was observed for entropy transfer with respect to the synaptic breakdown at the 9% level (Figure 7). Transfer entropy for CA3->CA1 at the synaptic breakdown level of 9% increased to 90% relative to the control. In addition to this, at the synaptic breakdown, the 35% level increased to 95% relative to the control model (Figure 7).

## 4. Discussion

The main symptoms of AD include memory disorders that are associated with the mechanism of recalling and memorizing new information, and in particular the impairment of memory functions in the hippocampus [2]. It is suspected that the disease develops gradually, but its etiology is not well understood so far [54]. Neuroanatomical research of the disease shows the disappearance of synaptic connections and, associated with this, the death of neurons. However, there are only computer simulations in agreement with hitherto statements about the staging of Alzheimer pathology in humans, especially around ‘preclinical’ stages ‘0’ and ‘1’ with very mild memory impairment, but with already existing degeneration of some stellate cells in layer II of entorhinal cortex in practically in all humans above 60 years of age.

Therefore, our microcircuit AD-like pathology model shows the dynamics of synaptic breakdown, modeling the gradual loss of synapses. The model is adequate for simulations of information flow through the hippocampus, but not for detailed pharmacological studies of AD treatment methods.

In the Horn–Ruppin model [3], the influence of synaptic connection breakdown on the functioning of memory was analyzed. The analysis showed that despite network damage, the introduction of a compensating factor that strengthens synaptic connections allows for proper functioning of memory within the network [3]. The extension of the aforementioned model was the examination of memory functioning using associative memory with the Hebb rule [5]. This model correctly simulated the change in the memory capacity of AD patients. The Hasselmo model showed the existence of an exponential increase of runaway synaptic modification, which is a mechanism of memory degradation in AD [6,7]. In this model, interference with previous information was discovered, which lead to a pathological increase in the strength of synaptic connections. The Menschik and Finkel models were based on the loss of cholinergic connections in AD [12,13,14]. These models were inspired by an exponential increase in the strength of synaptic connections, which preceded the loss of cholinergic connections. Diagnostic decision support of AD and analysis of SPECT images are the medical issues solved by means of artificial neural networks [55].

The aforementioned models focused on the analysis of a single mechanism in AD, without considering the neural parameters or describing hippocampus complexity in terms of analyzing chaotic behaviors and information theory. In our models, we showed a decrease in the number of spikes, spikes per burst and burst duration with an increase in synapse loss on granule cells of the dentate gyrus and on pyramidal neurons CA3 connections with EC2 and connections with EC2 on inhibitory interneurons (within DG and CA3). Our simulations have been carried out for both DG-CA3-CA1, CA3-CA1 as well as particular, individual regions. Our results are consistent with biological observations. A very interesting phenomenon is the simultaneous increase in inter-burst interval with a drop in the number of spikes, spikes per burst and burst duration in the dynamics of synaptic loss.

Analysis of nonlinear dynamics of simulation results in the pathological and the control model showed that the correlation dimension was significant in the DG-CA3-CA1 and DG regions. The complexity of the aforementioned regions was significantly higher at the synaptic breakdown level of 26% in relation to the control. A very interesting observation was the decrease of complexity of the hippocampus as a system at the synaptic breakdown level of 35% as compared with the control. It can be concluded that a larger loss of synapses is associated with a simpler network where a large number of freedom degrees of the analyzed system is not needed.

The information Shannon entropy is considered as the average amount of information increased in CA3-CA1, DG and CA1 regions. It is noteworthy that actually the linear entropy increases with the loss of synapses in CA1. The maximum entropy was obtained at the synaptic breakdown level of 35% in CA1, which determines the direction of time. The increase of Shannon entropy in the CA1 region indicates the irreversible process that is Alzheimer’s disease. According to the second law of thermodynamics, the increase in entropy is tantamount to a decrease in the available energy of the system. On the other hand, the increase in entropy is connected not only with the disorganization level, but also with the amount of information needed for description and complexity. Information in models with a synaptic breakdown at the 9% to 35% level is more complex compared with the control model.

Analysis of Lyapunov exponents showed that the control model of DG-CA3-CA1, CA3-CA1, CA3 and CA1 regions had a positive value; in other words, we have a stable system. A very interesting result was the positive Lyapunov exponents for all levels of synaptic breakdown. The pathological model was connected with system instability and chaos.

Information in the hippocampus flows in the direction of DG->CA3->CA1. The analysis of dynamics shows that Shannon entropy decreases in accordance with the direction of information flow. Information processes—the type of self-organization that occurs in the hippocampus and requires constant dissipation of energy—results in entropy decrease, which is associated with its increase for the entire brain. Very interesting results were obtained by analyzing the entropy transfer in the dynamics of synaptic loss. The values of synaptic breakdown at the 9% and 35% levels corresponded exactly to the change in the transfer of entropy in relation to the control model value in DG->CA3. The flow of information in DG->CA3 is closely related to the loss of synapses—weaker flow means greater synaptic breakdown. In contrast, in CA3->CA1 reverse relationship was received, where a larger synaptic loss was associated with a stronger flow of information.

## 5. Conclusions

The theory of information allows us to draw the conclusion that there is an interaction between the entorhinal cortex and DG-CA3 along with the loss of synapses. The coupling between DG and CA3 was stronger in the control model as compared with the pathological model. On the other hand, the interaction of the CA3 and CA1 regions indicate an inverse relationship. Our simulations of AD-like pathology using a simple but efficient hippocampal microcircuit model with a built-in long term synaptic potentiation (LTP) mechanism could provide further understanding of the clinical course of AD and could lead to new approaches of treatment for this disease.

## Figures and Tables

**Figure 1 entropy-21-00408-f001:**
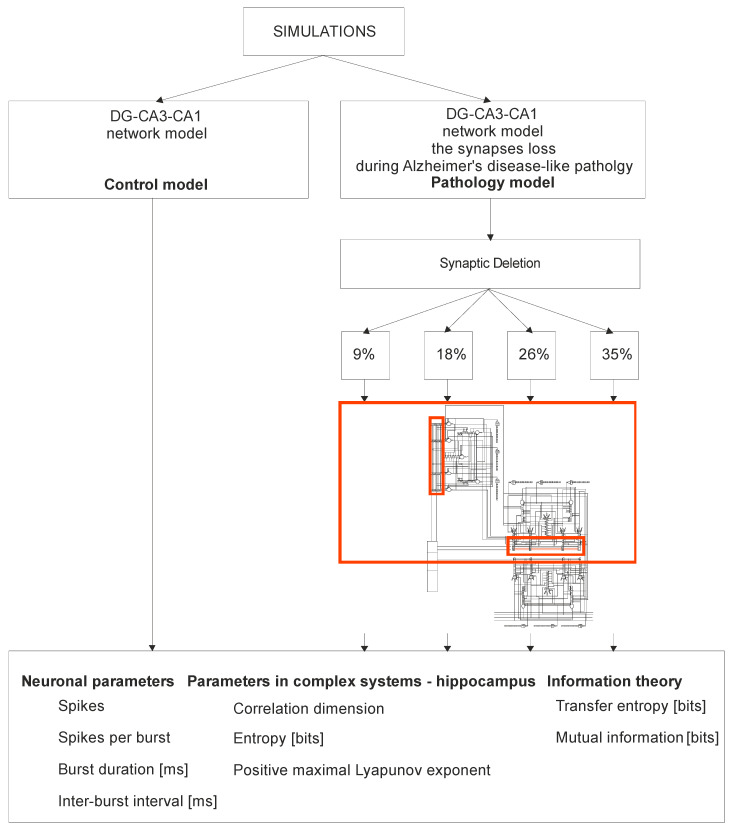
The diagram of the simulation of the hippocampus network DG-CA3-CA1 (control model vs. pathology model).

**Figure 2 entropy-21-00408-f002:**
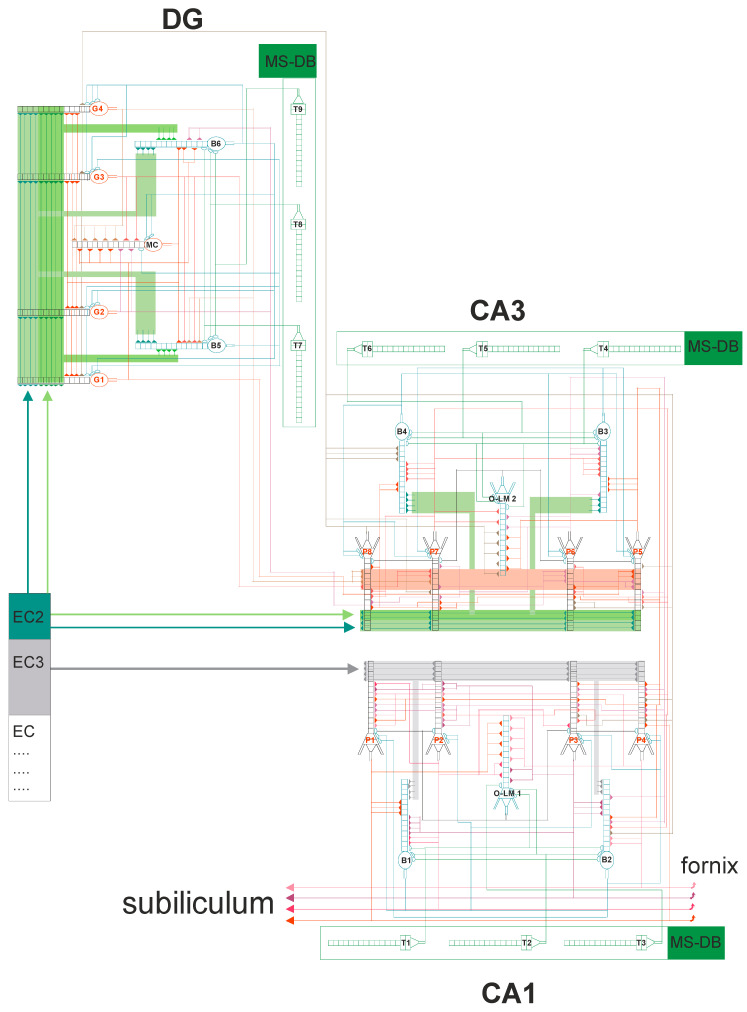
DG-CA3-CA1 hippocampal formation microcircuit. On the left is the dentate gyrus (DG) region, CA3 on the right and CA1 at bottom. Major cell types and their connectivity: granule (G1–G4), pyramidal (P1–P8), basket (B1–B6), OL-M (1–2) cells, mossy cell (MC). T1–T9 represent GABAergic cells in the medial septum-diagonal band (MS-DB) region which provides the disinhibitory inputs on hippocampal GABAergic interneurons at theta rhythm. All external and internal glutamatergic pathways innervate both principal cells (pyramidal and granule) and GABAergic interneurons (B and OL-M). The dentate gyrus and the CA3 area receive radially segregated layer II (EC2) inputs from both the medial (MEC) and lateral (LEC) entorhinal cortex. The combined and dentate-/CA3 processed MEC and LEC information is transmitted to CA1 pyramidal cells. Principal neurons in the entorhinal cortex layer 3 (EC3) directly project to the CA1 field and synapse with CA1 pyramidal neurons. The MEC and LEC are in reciprocal connections with different segments of the CA1 area and the subiculum, which however receive processed information from both MEC and LEC via the CA3 input.

**Figure 3 entropy-21-00408-f003:**
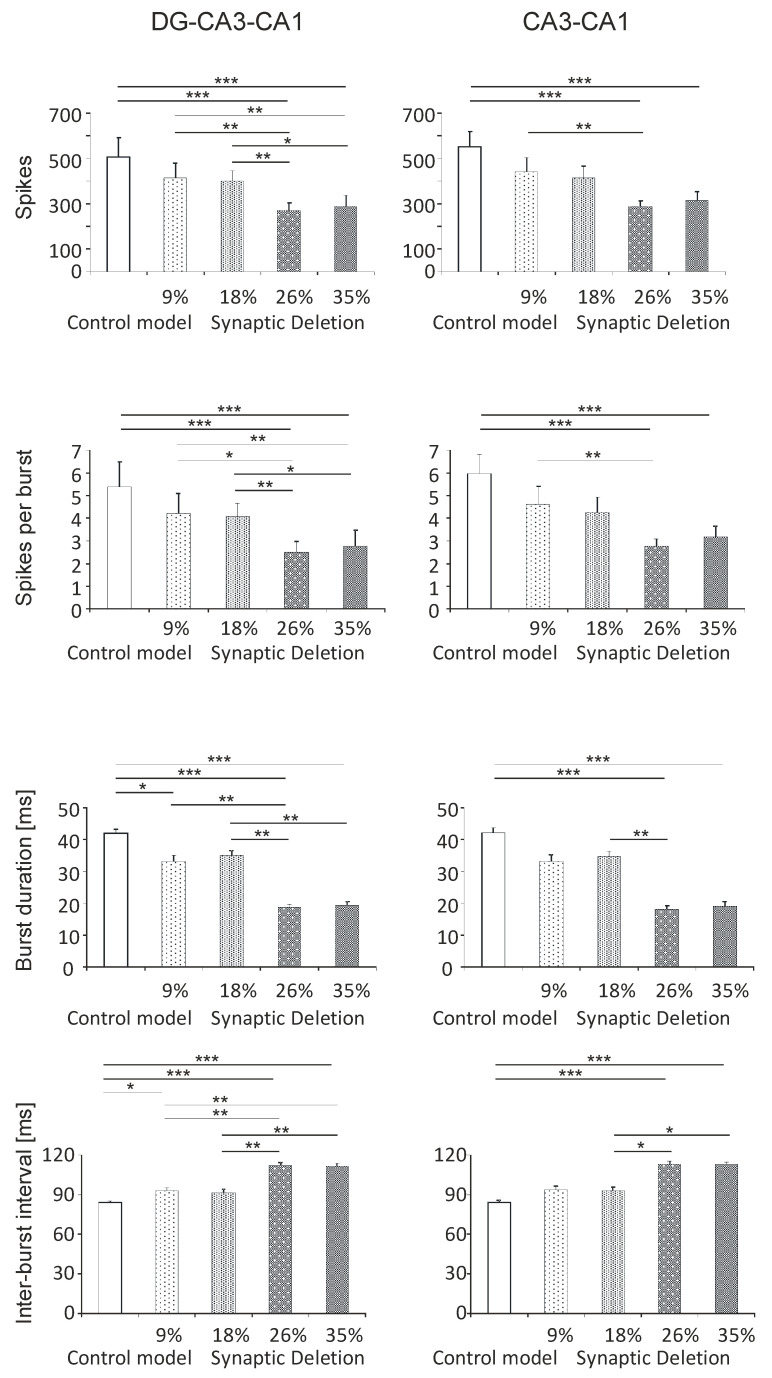
The analysis of simulations DG-CA3-CA1 region (left) and CA3-CA1 (right). The number of spikes, spikes per burst, burst duration and inter-burst interval for the pyramidal cell comparison control model and model of the synapses lost during an Alzheimer’s disease-like pathology: synaptic deletion 9%, 18%, 26% and 35% (* *P* < 0.05, ** *P* < 0.01, *** *P* < 0.001).

**Figure 4 entropy-21-00408-f004:**
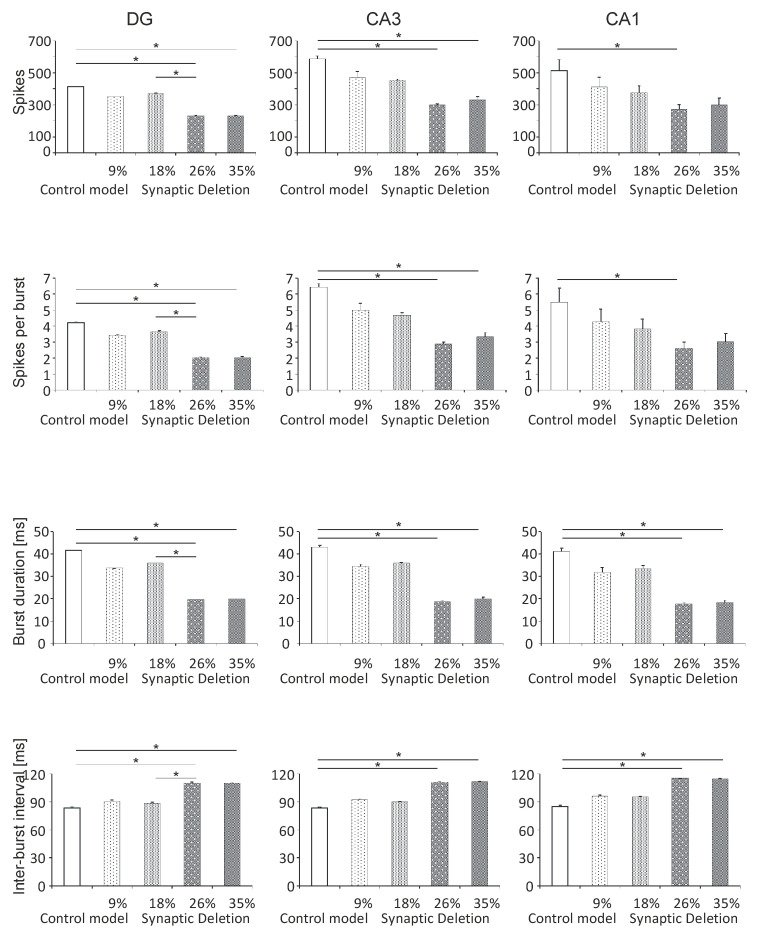
The analysis of simulations at the DG, CA3 and CA1 regions. The number of spikes, spikes per burst, burst duration and inter-burst interval for the pyramidal cell comparison control model and model of the synapses lost during an Alzheimer’s disease-like pathology: synaptic deletion 9%, 18%, 26% and 35% (* *P* < 0.05, ** *P* < 0.01, *** *P* < 0.001).

**Figure 5 entropy-21-00408-f005:**
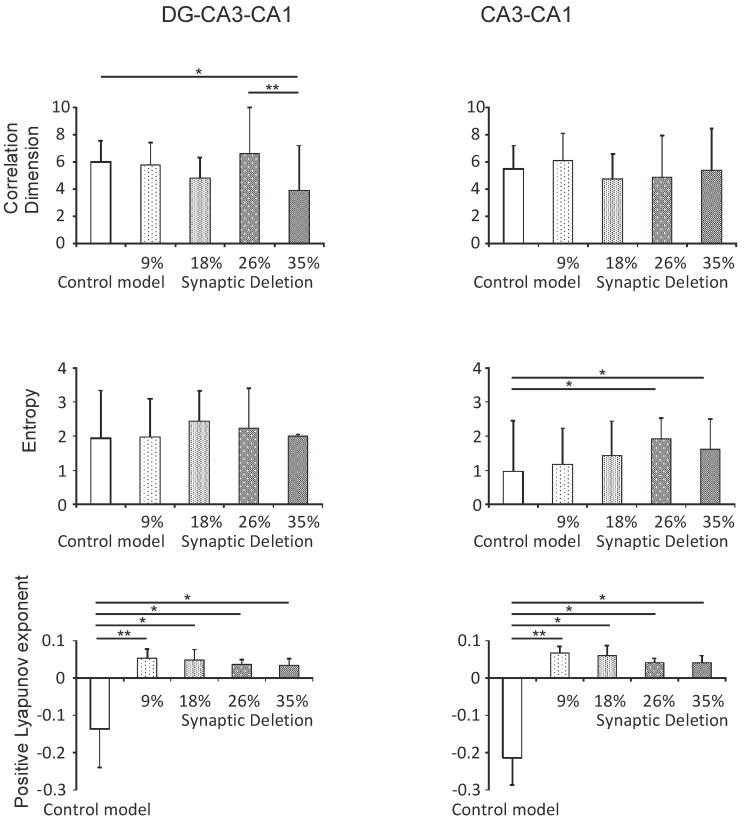
The nonlinear analysis of simulations at the DG-CA3-CA1 region (left) and CA3-CA1 (right). The comparison control model and model the synapses lost during an Alzheimer’s disease-like pathology: synaptic deletion 9%, 18%, 26% and 35% correlation dimension, entropy and positive Lyapunov exponent for the pyramidal cell comparison control model (* *P* < 0.05, ** *P* < 0.01, *** *P* < 0.001).

**Figure 6 entropy-21-00408-f006:**
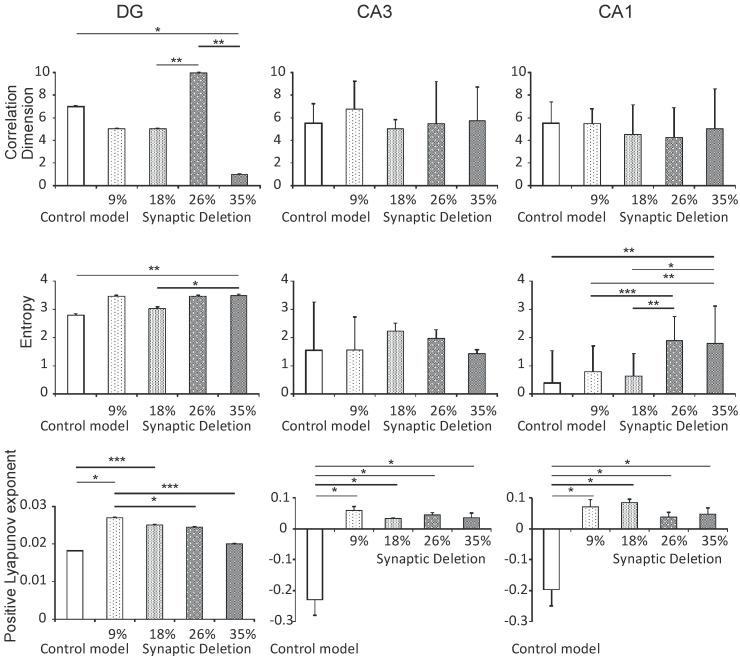
The nonlinear analysis of simulations at the DG, CA3 and CA1 regions. Comparison control model and model the synapses loss during an Alzheimer’s disease-like pathology: synaptic deletion 9%, 18%, 26% and 35% correlation dimension, entropy and positive Lyapunov exponent for the pyramidal cell comparison control model (* *P* < 0.05, ** *P* < 0.01, *** *P* < 0.001).

**Figure 7 entropy-21-00408-f007:**
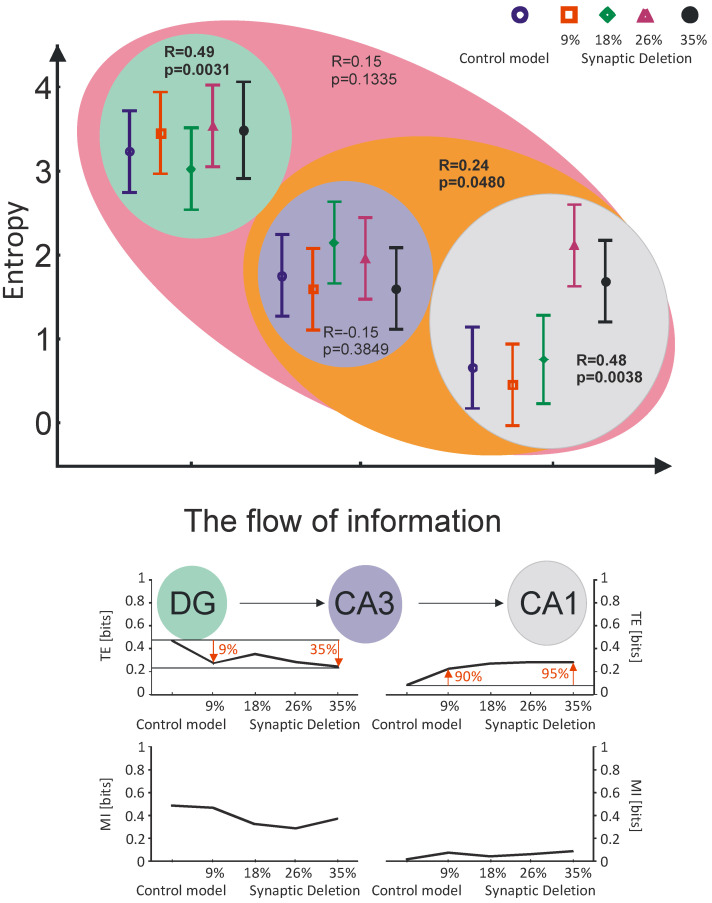
The flow of information analysis of the simulations at the DG, CA3 and CA1 regions. Comparison control model and model of the synapses lost during an Alzheimer’s disease-like pathology: synaptic deletion 9%, 18%, 26% and 35 transfer entropy (TE), mutual information (MI) for the pyramidal cell comparison control model.

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
