# Peer review of "Computer Model of Synapse Loss During an Alzheimer’s Disease-Like Pathology in Hippocampal Subregions DG, CA3 and CA1—The Way to Chaos and Information Transfer"

_entropy, 2019, doi:10.3390/e21040408_

Round 1
Reviewer 1 Report
In general, the manuscript is well written, only a few minor comments.
Consider checking the word Alzheimer disease and change by AD, there are several mistakes here.
Consider changing the first sentence of the material and methods sections 2.1 and 2.2. Do not start with figure sentence, add it later.
Consider revising the statistical methods, did you confirm if your data is parametric or non-parametric? Furthermore, describe ANOVA followed by post-hoc test, which post-hoc test? Also, consider change alpha by p-value.
Regarding the discussion section, consider adding the limitations of the study in order to contextualize your research.
Regarding the conclusion, please you can improve with two more conclusions that are really clear according to your results.
Author Response
In general, the manuscript is well written, only a few minor comments.
1. Consider checking the word Alzheimer disease and change by AD, there are several mistakes here.
It was be done
2. Consider changing the first sentence of the material and methods sections 2.1 and 2.2. Do not start with figure sentence, add it later.
Completion was added in red in lines 81-83.
Completion was added in red in lines 96-98.
3. Consider revising the statistical methods, did you confirm if your data is parametric or non-parametric? Furthermore, describe ANOVA followed by post-hoc test, which post-hoc test? Also, consider change alpha by p-value.
Completion was added in red in lines 160-167.
4. Regarding the discussion section, consider adding the limitations of the study in order to contextualize your research.
Completion was added in red in lines 382-390.
Neuroanatomical research of the disease show the disappearance of synaptic connections and associated with it the death of neurons. But there are only the computer simulations in agreement with hitherto statements about staging of Alzheimer pathology in humans , especially about “preclinical” stages “0” and “1” with very mild memory impairment, but with already existing degeneration of some stellate cells in layer II of entorhinal cortex, practically in all humans above 60 years of age.
Therefore, our microcircuit AD like pathology model shows the dynamics of synaptic breakdown, modeling the gradual loss of synapses. It is adequate for simulations of information flow through Hippocampus, but not for detailed pharmacological studies of AD treatment methods.
5. Regarding the conclusion, please you can improve with two more conclusions that are really clear according to your results.
Completion was added in red in lines 445-451.

Reviewer 2 Report
1. As authors state in the introduction section, their experience with simulations of the hippocampus network DG-CA3-CA1 is aimed at understanding the basic mechanisms of Alzheimer's disease and information flow in the hippocampus. Therefore, a short introduction about the "Information flow in the hippocampal–entorhinal loop" would set the scene for a wide audience favoring the understanding of the events described for the Alzheimer's disease-like pathology by the computer model (neural network) proposed. Authors should indicate that EC2 stands for entorhinal cortex layer 2 and DG for dentate gyrus.
As an example: Principal neurons in the entorhinal cortex layer 2 (EC2) project to the dentate gyrus (DG) and synapse on granule cells. EC2 principal neurons also project to the CA3 field and innervate CA3 pyramidal neurons. CA3 pyramidal neurons project to the CA1 and synapse with CA1 pyramidal neurons. Principal neurons in the entorhinal cortex layer 3 (EC3) directly project to the CA1 field and synapse with CA1 pyramidal neurons. CA1 pyramidal neurons send axons to deep layers of the EC.
2. In order to understand the complexity of the information flow, authors might explain how the changes in brain state affect EC and hippocampal neurons, differentially. This fact should be taken into account in order to reconcile whatever is going on under different brain states with the impairment of memory functions in the hippocampus.
For instance, brain-state-related firing rate changes of principal cells in upstream regions are not predictive of discharge rate changes in the downstream region. Moreover, brain-state-related firing rate changes of principal neurons and interneurons within the same layer (or region) are mostly uncorrelated, indicating that changes in brain state can drastically alter the balance between excitation and inhibition in a layer (or region)-specific manner.
Author Response
1. As authors state in the introduction section, their experience with simulations of the hippocampus network DG-CA3-CA1 is aimed at understanding the basic mechanisms of Alzheimer's disease and information flow in the hippocampus. Therefore, a short introduction about the "Information flow in the hippocampal–entorhinal loop" would set the scene for a wide audience favoring the understanding of the events described for the Alzheimer's disease-like pathology by the computer model (neural network) proposed. Authors should indicate that EC2 stands for entorhinal cortex layer 2 and DG for dentate gyrus.
As an example: Principal neurons in the entorhinal cortex layer 2 (EC2) project to the dentate gyrus (DG) and synapse on granule cells. EC2 principal neurons also project to the CA3 field and innervate CA3 pyramidal neurons. CA3 pyramidal neurons project to the CA1 and synapse with CA1 pyramidal neurons. Principal neurons in the entorhinal cortex layer 3 (EC3) directly project to the CA1 field and synapse with CA1 pyramidal neurons. CA1 pyramidal neurons send axons to deep layers of the EC.
We are in full agreement with above suggestions. Therefore we have substantially improved the caption for Fig. 2, which contains now all the abbreviations and main details about information flows in presented microcircuit. The caption should be the best site for explanations just below the diagram.
Completion was added in red in lines 107-117.
Figure 2. DG-CA3-CA1 hippocampal formation microcircuit. On the left Dentate Gyrus (DG) region, CA3 on the right and CA1 at bottom. Major cell types and their connectivity; granule (G1-G4), pyramidal (P1-P8), bascet (B1-B6), OL-M (1-2) cells, mossy cell (MC). T1-T9 mean GABAergic cells in Medial Septum – Diagonal Band (MS-DB) region which provides the disinhibitory inputs on hippocampal GABAergic interneurons at theta rhythm. All external and internal glutamatergic pathways innervate both principal cells (pyramidal and granule) and GABAergic interneurons (B and OLM). The dentate gyrus and the CA3 area receive radially segregated layer II (EC2) inputs from both the medial (MEC) and lateral (LEC) entorhinal cortex. The combined and dentate-/CA3 processed MEC and LEC information is transmitted to CA1 pyramidal cells. Principal neurons in the entorhinal cortex layer 3 (EC3) directly project to the CA1 field and synapse with CA1 pyramidal neurons. The MEC and LEC are in reciprocal connections with different segments of the CA1 area and the subiculum, which however receive processed information from both MEC and LEC via the CA3 input.
2. In order to understand the complexity of the information flow, authors might explain how the changes in brain state affect EC and hippocampal neurons, differentially. This fact should be taken into account in order to reconcile whatever is going on under different brain states with the impairment of memory functions in the hippocampus.
For instance, brain-state-related firing rate changes of principal cells in upstream regions are not predictive of discharge rate changes in the downstream region. Moreover, brain-state-related firing rate changes of principal neurons and interneurons within the same layer (or region) are mostly uncorrelated, indicating that changes in brain state can drastically alter the balance between excitation and inhibition in a layer (or region)-specific manner.
We fully agree that changes in brain state can drastically alter the balance between excitation and inhibition in a layer (or region)-specific manner, as showed in:
1. Miyawaki H, Diba K. Regulation of Hippocampal Firing by Network Oscillations during Sleep. Curr Biol. 2016 Apr 4;26(7):893-902. doi: 10.1016/j.cub.2016.02.024.
2. Mizuseki K, Miyawaki H. Hippocampal information processing across sleep/wake cycles. Neurosci Res. 2017 May;118:30-47. doi: 10.1016/j.neures.2017.04.018.
But there are only the computer simulations in agreement with hitherto statements about staging of Alzheimer pathology in humans , especially about “preclinical” stages “0” and “1” with very mild memory impairment, but with already existing degeneration of some stellate cells in layer II of entorhinal cortex, practically in all humans above 60 years live. We have not performed EEG experiments. And at microcircuits level the most significant values to evaluate are ability to coincidence detection , time coding of output spike trains and ability to learning according to Long Term synaptic Potentiation (LTP). In presented microcircuit we have 182 excitatory synapses with LTP (on 8 pyramidal, 4 granule and 2 OLM cells), in technical terms 182 independent analog memory micro devices, the each value must be newly calculated in each simulation step at 0,5 millisecond. Thus we do not use the firing rate evaluations. Below an example of raster plot of principal cells firing during 10 seconds of simulation – color changes mean shifting of particular spike trains on time axis.
Completion was added in red in lines 382-390.
